

# Reconfigurable monitoring for telecommunication networks

Man Tianxing[1], Vasiliy Yurievich Osipov[2], Alexander Ivanovich Vodyaho[3], Andrey Kalmatskiy[4], Natalia Alexandrovna Zhukova[2], Sergey Vyacheslavovich Lebedev[3] and Yulia Alexandrovna Shichkina[3]

[1] ITMO University, Saint Petersburg, Saint Petersburg, Russia
[2] Saint-Petersburg Institute for Informatics and Automation of the Russian Academy of Sciences, Saint Petersburg, Saint Petersburg, Russia
[3] St. Petersburg State Electrotechnical University, Saint Petersburg, Saint Petersburg, Russia
[4] Google, New York, NY, United States of America

## ABSTRACT

This article addresses the monitoring problem of the telecommunication networks. We consider these networks as multilevel dynamic objects. It shows that reconfigurable systems are necessary for their monitoring process in real life. We implement the reconfiguration abilities of the systems through the synthesis of monitoring programs and their execution in the monitoring systems and on the end-user devices. This article presents a new method for the synthesis of monitoring programs and develops a new language to describe the monitoring programs. The programs are translated into binary format and executed by the virtual machines installed on the elements of the networks. We present an example of the program synthesis for real distributed networks monitoring at last.

Corresponding author
Man Tianxing, mantx626@gmail.com

# INTRODUCTION

Nowadays, the success of human activity in many areas largely depends on the efficiency of the functioning of telecommunication networks. Telecommunications networks today are used in a variety of fields of human activity. Thus, the use of radar and radar-optical data to solve a massive class of problems in the field of transport, agriculture, classification of area and point objects, monitoring of anthropogenic activities, and other tasks seem to be extremely promising. At least a dozen of new radar satellites are planned to be launched in the coming years, all cars today are equipped with radar. The intelligent processing of this data requires the use of edge and cloud computing, which entails the new challenges to the telecommunication network design.

Damage or incorrect operation of such networks can be a reason for severe losses. So, it is necessary to have adequate tools and methods for maintaining such networks. It can be done through permanent monitoring. Monitoring ensures safety operation, quick detection, and prevention of network failures. Many factors cause changes in the structure, the state, and the behavior of the networks. Thus, there is a need to reconfigure monitoring

processes. Reconfigurable monitoring processes should allow us to gather the data about dynamic telecommunication networks and process them according to the current state of these networks and estimations of their expected use.

In the telecommunication networks, objects of monitoring are multiple technical devices of end-users. Sets of parameters characterize the state of the objects and their elements. Centralized and decentralized schemes for object monitoring are used. Centralized schemes assume that monitoring systems gather data from objects. Also, data can be gathered by the objects and transferred to the monitoring systems. In both schemes, predefined processes are commonly used. Gathering all possible data leads to a significant increase in network traffic. When data is partially collected, it does not allow quickly to detect failures of the objects, identify their reasons.

For the reconfiguration of monitoring processes, business rules can be used. The rules define processes behavior for different known situations. This approach is well applicable for monitoring when networks have permanent structures and permanent conditions of use. The problem of monitoring of dynamic networks remains unsolved. To solve the problem monitoring systems must have abilities to build and rebuild the monitoring processes dynamically.

In the article, we propose a new method for building monitoring processes based on using the theory of synthesis. To execute the processes a new language and a new virtual machine (VM) are developed.

The rest of the article is structured as follows. In the second section, the capabilities of the existing monitoring systems are analyzed. It is shown that for dynamic telecommunication networks reconfigurable systems have not been considered until now. Then we describe monitoring processes and propose new models for monitoring programs and the method for their synthesis that allow develop reconfigurable monitoring systems. In the fourth section we present the implementation of the proposed method, which contains the description of the language for monitoring programs, a compiler for the language, and a virtual machine that is capable of executing the compiled monitoring programs on end-user devices. In the fifth section, a case study is given to prove the effectiveness of the method. The last section is the conclusion.

## MATERIALS & METHODS

### Related work

Today monitoring systems are used almost in all subject areas –medical (*Baig et al., 2017*; *Mukhopadhyay, 2014*; *Sohn et al., 2003*; *Yang et al., 2016*; *Shichkina et al., 2019*), environment (*Gaikwad & Mistry, 2015*; *Kumar, Kim & Hancke, 2012*), social network (*Chakravarthi, Tiwari & Handa, 2015*), energy (*Eissa, Elmesalawy & Hadhoud, 2015*), production (*Singh, Gernaey & Gani, 2010*), urban, economy (*Bello et al., 2018*; *Sakhardande, Hanagal & Kulkarni, 2016*), agriculture (*Suma et al., 2017*), transport (*Shichkina et al., 2019*), fishery (*Raju & Varma, 2017*), etc. For many subject areas, including telecommunication, monitoring systems are playing a critical role.

In traditional monitoring systems (Zabbix, http://www.zabbix.com/), (Nagios, http://www.nagios.org/), (Zenoss, http://www.zenoss.Org/), (Pandora FMS https:

//pandorafms.com/), (Icinga https://icinga.com/), (Sensu http://sensu.io/), (GroundWork Monitor https://www.gwos.com/), data is collected and then transmitted to monitoring systems in the form of the data streams. The data is processed by monitoring systems which have a distributed tree structure. Each element can collect and process data from multiple objects. The higher-level elements aggregate the data processing results of the lower-level elements. The tree structure allows the monitoring systems to have a sufficiently high level of performance. If the observed object changes and data is generated, redesign of the monitoring systems is a necessary process. Systems redesign assumes active involvement of software architects and programmers.

In the past few years, the monitoring systems have evolved significantly. Developed monitoring systems are based on the integration of multiple modern technologies (*Morgan & O'Donnell, 2018*). In particular, in these systems, data is collected using Internet of Things technologies (*Maksymyuk et al., 2017*; *Myint, Gopal & Aung, 2017*; *Raju & Varma, 2017*; *Sakhardande, Hanagal & Kulkarni, 2016*; *Suma et al., 2017*; *Yang et al., 2016*). Data processing employs statistical methods, data mining algorithms, and machine learning algorithms (*Ge, Song & Gao, 2013*). These monitoring systems are a type of adaptive systems. They assume changes in both data gathering processes and processes of data processing. Adaptive systems use flexible data processing models (*Cabal-Yepez et al., 2012*; *Dovžan, Logar & Škrjanc, 2014*) or business rules (*Al Mamun et al., 2016*; *Koetter & Kochanowski, 2015*) to build and rebuild processes. The likelihood of their ability to adapt in different cases depends on the composition of the selected business rules and the defined values of customizable parameters of the processing models (*Singh, Gernaey & Gani, 2010*).

The reconfigurable monitoring systems have advanced capabilities for adaptive data gathering and processing. The features of reconfigurable systems are discussed in (*Korableva, Kalimullina & Kurbanova, 2017*; *Lyke et al., 2015*). In monitoring systems, reconfiguration operations can be considered at several levels: sensor level, embedded system level, and application level. At the sensor level, reconfiguration problems are solved by developing new programmable sensors (*Myint, Gopal & Aung, 2017*). The new programs are uploaded for updating the logic of the sensors. For the adaptation of the embedded monitoring systems, they equip with reconfigurable data preprocessing modules (*Cabal-Yepez et al., 2012*). At the application level, the adaptability is achieved through customizable interfaces (*Fu, 2012*), usage of genetic algorithms (*Thompson, 2012*), and so on. It is also possible to synthesize new configurations for the systems and their components at the level of field-programmable gate arrays (*Cong et al., 2011*; *Cong & Minkovich, 2007*; *Lysecky, Vahid & Tan, 2004*; *Rubin & DeHon, 2011*).

For implementing reconfigurable systems, several methods have been proposed. In (*Morgan & O'Donell, 2014*), service-oriented architectures of reconfigurable systems are presented. The problems of message routing and service orchestration in such systems are discussed. Technological solutions for building reconfigurable systems are given in (*Silva, Marques & Lopes, 2018*).

One of the limitations of the existing systems is that they cannot reconfigure themselves dynamically. For dynamic reconfiguration, they have to be able to synthesize programs

 

for data gathering and data processing and execute them. Some of the processes should be performed in the monitoring systems and some of the end-user devices.

To date, only a few dynamically reconfigurable systems exist. One of such systems has been developed for monitoring the status and performance of a supercomputer network (*Stefanov et al., 2015*). The monitoring system is considered as a set of subsystems. The subsystems process data about the performance of one or more tasks. Performance metrics are calculated on the fly. To this purpose, a program module is dynamically created for each process that is performed to solve the general task, and the transmission paths for the calculated metrics are dynamically determined. The module oriented architecture of the systems and their ability to create the modules dynamically make them able to adapt to new data sources and new processing methods. When creating a new module, the nodes on which it will be executed are selected based on the current load on the computer network. As a result, a balanced load of the nodes is ensured. The nodes of the system can process a part of the data about the state of the network. With these additional resources for solving the monitoring problems are obtained. The results of using these monitoring systems have shown that they allow solving the tasks of monitoring of supercomputer networks.Reconfigurable systems and dynamically reconfigurable systems have not yet been developed in the field of telecommunications.

# MULTILEVEL SYNTHESIS OF MONITORING PROGRAMS

## Monitoring objects, processes, and programs in telecommunication networks

Telecommunication networks have a multilevel structure. The elements of the higher levels are groups of objects. Within the lower levels, separate objects are considered. These objects are end-user devices. The components and the subcomponents inside characterize them. There are multiple links between the elements at all levels.

Monitoring processes are oriented on providing data about the state of the end-users' devices to the customer service. These devices can gather data about their state and process it. Monitoring systems can interact with the devices. Within the interaction, data can be received from the devices, and commands on data gathering and data processing can be sent to them. The devices can perceive both separate commands and programs that contain multiple commands.

Monitoring programs are built and rebuild by monitoring systems. New programs are built-in multiple cases, such as changes in the structure, state, or behavior of the devices or definitions of new requirements to the results of the monitoring. Requirements can refer to both the output data about the state of the network and the conditions of the monitoring, for example, admissible time modeling, accuracy and reliability of the models, consumed technical resources, etc.

There are two primary sources of data about the state of the devices. They are the messengers that are sent from the devices and the log files that they produce. The log files are sequences of records that describe events registered by the devices. Related data on factors affecting the network operation can be received from other sources.

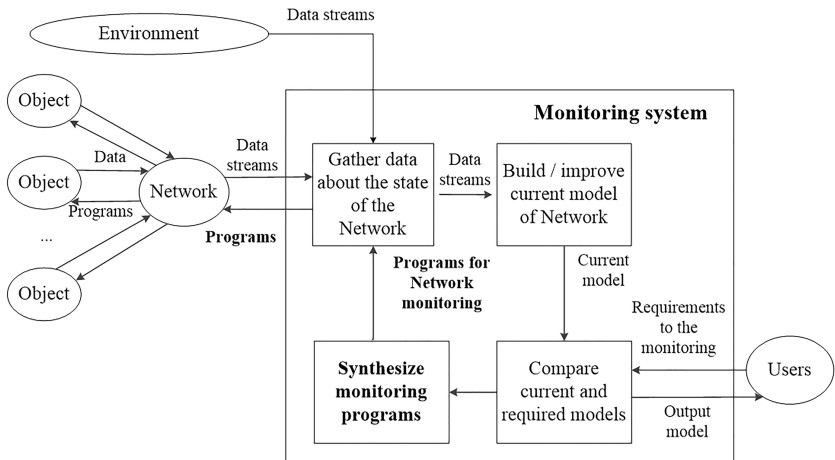

**Figure 1  The structure of the monitoring process.**

The operators of the customer service define the requirements for the results of the monitoring. Figure 1 represents the structure of the monitoring process.

According to Fig. 1, monitoring processes assume gathering data about the target objects (end-user devices). Current models of target objects are built using gathered data. These models reflect the state of the objects based on available data. Current models are compared with the required models. They are defined according to the requirements of the end-users (operators of customer service). Usually, these two models do not match. Current models can have many undefined parameters in comparison with the required models. If the models do not match, then new monitoring programs are synthesized. Possible ways of transitions between current and required models are considered to synthesize new programs. Each transition determines additional elements of data that are needed to reach the target model. Thus, after constructing sequences of transitions, it becomes known what data should be gathered. New monitoring programs contain sequences of commands that allow gathering required data elements.

The formally monitoring process $PR$ can be defined as $PR : O' \rightarrow O''$ where $O'$ is the model that reflects the state of the objects in the network. The model $O'$ is the current model that is build using gathered data. The model $O''$ is the model that is required by the user. The monitoring processes are targeted at building current models that match the required models.

The synthesis of monitoring programs is an essential part of the monitoring processes. Synthesized programs are defined as the complement of $O''$ concerning $O'$: $PRG = O'' \setminus O' = \{e \in O'' | e \notin O'\}$.

The synthesis of the monitoring programs is ensured through the following:

• The networks are commonly described in discrete time and discrete space. Thus automata are used as the models of the networks. Automata describe the states of the networks. Since there are too many possible states, some of them can be unknown, it is not possible to describe all the states of the automata apriori. Due to that relatively finite

automata are used (*Osipov, 2016*). They are finite only for one moment of time. At each time step, all the parameters of the automata can be redefined.

- Traditional methods of synthesis have high computational complexity. The complexity can be reduced if we take into account the multilevel structure of the networks. To synthesize models of objects with multilevel structure multilevel methods of synthesis can be used. Compared to single-level methods, they allow increase the speed of the synthesis of the required programs in times. It is achieved due to the fact that the number of the elements in the models at upper levels is small in relation to the lower levels.
- The synthesized programs can be described using a new language. It is oriented on describing the programs for monitoring of telecommunication networks. The programs that are written using this language can be compiled and executed on end-user devices.

## Automata models for programs synthesis

The models of the objects are multilevel structures that represent the objects as multiple sets of parameters. Sets of parameters and separate parameters are linked within separate levels and between the levels.

We suggest using multilevel automata models to synthesize the programs of monitoring for the objects that assume gathering information at different levels.

The model of the automata at one level can be defined in the following way (*Osipov, 2016*; *Osipov et al., 2017*):

$< AI, AO, AS, AR>$, where AI - a set of admissible input data, AO - a set of admissible output data, AS - a set of admissible internal states, AR - an admissible set of transition functions between the elements of the set of input data and the elements of the set of output data.

So, the entire multilevel model is

$< < AI^0, AO^0, AS^0, AR^0 > ; < AI^1, AO^1, AS^1, AR^1 > ; ..... < AI^N, AO^N, AS^N, AR^N > >$, where the super index defines the level of the automata model.

At each step of program synthesis, automata can be in one of the admissible states. Within each step, the sets of admissible states and transition functions are rebuilt, taking into account the achieved state and the conditions of synthesis. Thus the automata at step r is defined as: $< I_r, O_r, S_r, F_r, AI_{r-1}, AO_{r-1}, AS_{r-1}, AR_{r-1} >$ , where I - input data, O - output data, S - internal state, F - transition functions (conditions). Parameters I, O, S, R are defined for the step r and parameters AI, AO, AS, AR - for the step r-1.

## Method of synthesis of monitoring programs

The method allows synthesizing programs for objects monitoring in conditions when the current and the target models of the objects are defined. It is required to gather data about the objects that are necessary to move towards required models from current models.

The method uses direct and backward inference. The direct inference is used to prove that the transition between models is possible and that the target model exists. As a result of the direct proof, multiple sequences of transitions can be defined. The backward inference allows restoring programs from the results of the proof. The inference is carried out according to the scheme mirrored to direct inference.

In the conditions, when the existence of the target models cannot be proved, the possibilities of changing the models can be considered.

In the automata model for program synthesis, the set of input data elements $\{d_s\}$ are defined according to the current model of the object, the set of output data elements $\{d_w\}$ are elements of the target model of the object. Transition functions $F_{zv}$ reflect links between elements of the models: $F_{zv}(d_{zve}) \rightarrow (d_{zva})$, where $e = [1, E_z]$ indicates elements of the models that are required to achieve an element indicated as $d_{zva}$.

## Method for building sequences of transitions between object models based on direct inference

Below the description of the method is given:

*Input data.* $\{d_s\}$ - the sets of the input data elements; $\{d_w\}$ - the sets of the output data elements; $F_{zv}$ - transition functions between the elements.

*Output data.* TransitionSequence - sequences of transitions that allow reach elements $\{d_w\}$ of the target model having elements $\{d_s\}$ of the current model.

The pseudo code of the method is the following:

Function FindAchievableElements(maxLevel, M, CurrentM, TargetM) is

```
1.   for each l in [maxLevel..0] do
2.     for each i in [0..M.funConditions[l].length] do
3.       if checkCondition(M.funConditions[l][i]) == true do
4.         elementPairs = cartesianProduct(CurrentM.Elements[l], TargetM.Elements[l])
5.         for each j in [0..elementPairs.length]
6.           if profTransition(M.vzLinkingFun[l][i](elementPairs[j].first, elementPairs[j].second)=true do
7.             M.AchievedFrom[l] = add(M.AchievedFrom[l], elementPairs[j].first)
8.             M.AchievedTo[l] = add(M.AchievedTo[l], elementPairs[j].first)
9.             M.AchievedWith[l] = add(M.AchievedWith[l], M.vzLinkingFun[l][i])
10.            M.AchivedWhile[l] = add(M.AchivedWhile[l], checkCondition[l][i])
11.            remove(M.NonProven[l+1], elementPairs[j].second)
12.          else
13.            M.NonProven[l]= add(M.NonProven[l], elementPairs[j].second)
14.        end
15.      if M.NonProven[l].length != 0 && maxLevel != 0 do
16.        FindAchievableElements(maxLevel-1, M, CurrentM, TargetM)
17.    end
18.  return M
```

Function DirectProgramSynthesis(maxLevel, M, CurrentM, TargetM) is

1. *M = FindAchievableElements(maxLevel, M, CurrentM, TargetM)*
2. *if M.NonProven.length == 0 do*
3.    *for each l in [0..maxLevel] do*
4.      *M.TransitionSequence = zip(zip(M.AchievedWith[l], M.AchievedWhile[l]), zip(M.AchievedFrom[l], M.AchievedTo[l])) /\* to get a list of quads of the form (function, condition, input, output)\*/*
5. *else error 'no transition is found'*
6. *return M*

## Method for restoring programs from the results of the proof based on backward inference

Below the description of the method is given.

*Input data.* {dw} - the sets of the input data elements; {ds} - the sets of the output data elements. Target data elements are considered as the input, and the input data elements as the target elements because the inference is backward.

*Output data.* PRG -models of monitoring programs.

The pseudo-code of the method is the following:

Function BackwrdProgramSynthesis(maxLevel, M, CurrentM, TargetM) is

The resulting program is a sequence of program structure elements that can be converted to program code written using scripting languages.

## Complexity of multilevel synthesis of monitoring programs

The upper boundary of time $T_H$ required for program synthesis using the proposed method can be estimated by the following formula:

$$T_H \approx c \sum_{i=0}^{K} m_i^2 \leq c (\sum_{i=0}^{K} m_i)^2 \tag{1}$$

where: c –constant-coefficient; $m_i$- the number of conditions on i-th level. It is necessary to mention that $m_i$ is essentially less than the total number of the conditions of the automata model.

Taking into account that on upper levels each inference step is equivalent to $n_i$ steps at the level "0", one can get a lower boundary of time $T_L$ of multilevel synthesis of the programs,

$$T_L \approx c \sum_{i=0}^{K} \frac{m_i^2}{n_i^2} \leq c \sum_{i=0}^{K} m_i^2 \tag{2}$$

The average estimate of time T of multilevel synthesis concerning (1), (2) is equal to $T = (T_L + T_H)/2$.

1.  *PRG = initializeProgram*
2.  *MProc = initializeProgramModel*
3.  */* Build the model of monitoring program using backward inference, TargetM and CurrentM are swapped*/*
4.  *MProc = DirectProgramSynthesis(maxLevel, MProc, TargetM, CurrentM)*
5.  *MProg = reproduceTransitionSequences(MProc)*
6.  */* Build the model of the monitoring program according to the synthesized structure */*
7.  *for each l in [(maxLevel-1)..0] do*
8.      *for each i in [0.. MProc.AchievedWith[l].length] do*
9.          *z = getFunKind(MProc.AchievedWith[l][i])*
10.         *v = getFunType(MProc.AchievedWith[l][i])*
11.         *MProg.StructureElements[l] = add(MProg.StructureElements[l], buildStructureElement(z, v))*
12.     *End*
13.     *for each i in [0.. MProc.AchivedWhile[l].length] do*
14.         *conditions = MProc.AchivedWhile[l][i]*
15.         *MProg.Condition[l] = add(MProg.Condition[l], buildConditions(conditions))*
16.     *end*
17.     *MProg.Program[l] = buildProgramForLevel(MProg.StructureElements[l], MProg.Condition[l])*
18.     */* transform program structure elements of each level to the resulting program structure */*
19.     *PRG = buildProgram(PRG, MProg.Program[l])*
20. *end*
21. *return PRG*

## IMPLEMENTATION OF MONITORING PROGRAMS

### Language for monitoring programs

We design the language to describe the programs for monitoring of telecommunication networks. The programs are intended for execution on end-user devices.

The following requirements were imposed on the capabilities of the language. The language should allow configurable filtering and data aggregation parameters, determine the composition of messages sent from the devices, specify commands for performing actions by the devices, and determine the conditions for their execution.

Vocabulary and syntax are defined for the modeling language. The vocabulary is presented in Table 1, the syntax is given as follow:

*program=:'module' id ';' (global_var_definition | function_definition)\* < end of file>*
*global_var_definition=:id(name) '=' expression ';'*
*function_definition=:id(name) '(' (id(param_name) (',' id(param_name))\*)? ')' '{' (statement)\* '}'*
*statement =:expression ';'*
*| id(local_var) '=' expression ';'*
*| 'if' '(' expression ')' statement ('else' statement)?*
*| (id(label) ':')? 'while' '(' expression ')' statement*
*| (id(label) ':')? 'do' statement 'while' '(' expression ')' ';'*

**Table 1   Vocabulary of the modeling language.**

| Symbol | Desription |
|---|---|
| space, tab, \n, \r | separator |
| // or /* */ | comment |
| [A-Za-z_][0-9A-Za-z_]* | identifier |
| \" \n \r \t \xNN \\\0" | literal string |
| {} | block of code |

 | (id(label) ':')? 'for' '(' initializer_list ';' expression ';' (expression (',' expression)*)? ')'

statement
 | '{' (statement)* '}'
 | 'break;'
 | 'return' expression ';'
exression =:ternary
ternary =:concatenation ('?' ternary ':' ternary )?
concatenation = logical ('_' logical)*
logical =:comparison (('&&' | '||') comparison)*
comparison =:math (('< ' | '< =' | '> ' | '> =' | '==' | '!=') math)?
math =:muls (('+' | '-' muls)*
muls =:unary (('*' | '/' | '%' | '*' | '&' | '|' | '< < ' | '> > ') unary)*
unary =:unary_head unary_tail
unary_head=:NUMERIC_CONST | STRING_LITERAL
 | id(var_name)
 | ('-' | '~' | '!' | '++' | '−') unary_head
 | '(' expression ')'
 | '[' id(field_name) (',' id(field_name))* ']' (('*' muls) / ('{' expression (',' expression)*
'}'))?
'
unary_tail=:'[' expression (':' expression)? )']'
 | (':=' | '+=' | '-=' | '/=' | '_=' ....) expression
 | '(' expression (',' expression)* ')'
 | '.' id(field_name)
 | '++' | '−'

The source codes of the programs are encoded in utf8.

The modeling language is a typed language. Types are not converted to each other automatically. Each expression has a type known at the compilation stage. Types have no names. The same types are those that have the same constructors and the same sets of parameters.

The language supports the following data types: int (32 bits with a sign), bool (false | true), string (characters in utf-8, addressed byte by byte), table (an array of variable size, containing structures with fields), link to a function, array of variable size.

Variables in the language are set in the form: "name = initializer". The type of initializer simultaneously sets the type of the variable. It is forbidden to declare an uninitialized

variable. Variables starting with "_" are reserved for platform variable names (for example, _currentTime). A variable that has not been modified in the program code are considered as constant when compiled.

Global variables are initialized before the function that starts the program is called. They are initialized in the order of their declaration in the program. It is possible to access global variables before declaring them. If during the initialization of global variables functions are called, then in these functions, global variables that are not yet initialized cannot be used.

Changes in variables (assignment) is performed by a group of operators + =, & = b, etc. To assign a new value to a variable, the operator:: = is used.

Local variables are defined in context from the declaration point to the end of the block in which they are declared. Local variables declared in the header of the for loop are defined in this header and the loop body.

Declaring two identical variables in the same context is not allowed. Also, overriding (hiding) an external context variable is not allowed.

The language provides the following control structures: block of operators, declaration of a local variable, expression, condition, loop, exit from the loop, exit from the function, and return value.

The functions in the proposed modeling language are described in the form: 'name (parameters) {actions}'. Nested local functions are not supported, as this leads to the need to work with closures and increases the complexity of the program.

The result of the return expression determines the type of result. The types of parameters are determined by the actual parameters passed in the first function call. The types of parameters in all calls must match because using polymorphism or instantiating templates can lead to an increase in the size of the generated code and to increase the time of program execution.

Functions can be called before they are declared. Functions can be called directly, assigned to variables, passed as parameters, and returned as results.

When working with strings, the following operations can be performed: assigning a literal constant, concatenation, obtaining the length of a string, converting a string to a number, converting a number to a string, receiving a substring, comparing strings, parsing a string and patterns, searching for a string in a table, parsing parameters.

The following options are provided for working with the tables: create a table, create a table with initialization, access fields, get the number of records, insert records, delete records, search in the table by a numeric key, search in the table by a string key.

When working with arrays, it is possible to create them, create with initialization, access elements, get the size, insert elements, delete elements, search for an element.

The data is encoded by the int data type and corresponds to Unix time.

Further extension of the developed language is possible, in particular, support of program interfaces, class structures, inheritance mechanisms.

## Program models for monitoring

The program is translated into a binary format. The translation is provided through a developed compiler. As a result of the translation, a program module is formed. During

the conversion, the compiler checks the syntax of the program text, the consistency of the names, and the compatibility of the type.

When describing a program in binary format, instructions in the general command language are used, which makes it possible to extend the developed language. Instructions are strictly typed that allows high-speed program execution. The compactness of the binary programs is ensured by using language structures that are close to the solved monitoring tasks and by the compactness of the structures themselves.

Monitoring programs are intended for execution by the virtual machines. Loading and unloading modules, their registration and calls assumed to be carried out through the software interface, which is provided by the machines:

*int vm_load_module (void\* data); // data should be allocated with vm_alloc*
*void vm_free_module (int id);*
*size_t vm_lookup (char\* fn_name);*
*void vm_call (size_t ord, char\* params);*
*void \*vm_alloc (size_t size);*

Modules cannot call each other's functions or exchange information.

When the module is loaded, the start () function that registers the named event handlers is executed.

The predefined handlers are the following:

"t" –a timer handler that is called once a second in the idle mode;

"c" –a handler that is called in response to a console command;

"s" –a handler that is called, before sending any packet, allows adding the necessary fields and events to the packet.

Parameters are passed to the handlers as text strings with separators "|". In the language, the operators that allow extract parameters from the strings of this format are supported.

*// if called myHandler("123|Hello");*
*myHandler(p)*
*{*
*screen = intArg(); // screen = 123*
*mode = strArg(); // mode = "Hello";*
*...*
*}*

The loadable modules have access to the sets of system variables, parameters provided by system functions. They can initiate actions that can be performed by the system functions.

*TimerProc(s) {*
*if (_systemMemFree < 1024\*1024)*
*reboot();*
*}*

When executing code in a virtual machine, the following verification procedures are performed: availability of free memory, stack overflow error, looping, index out of array bounds error, accessing uninitialized fields, division by zero, getting the remainder of division by zero. When an exception occurs, the module is unloaded, and its resources are freed, then its handler stops perceiving information about external events.

## A virtual machine for monitoring

The synthesized monitoring programs are executed by the virtual machine (VM) [29], whose structure is shown in Fig. 2.

The main elements of the virtual machine are the following:

CN (Computer Network) Agent - an application running on a network element and containing a virtual machine (CN VM) in which program modules are executed;

CN VM (Virtual Machine) –the environment for the program modules execution;

CM VM prodModule - an agent which logic is determined by the synthesized programs that are loaded on the elements of the networks;

Logger –an element of the network that is responsible for collecting and providing log files produced by the elements of the networks.

The logic of using the virtual machine is as follows. Program modules are compiled for each of the synthesized programs. The modules are placed on the Data Provider. When new modules appear, they are loaded into the VM. To load the modules, DTS is used. VM ensures the execution of the modules. The logic of data gathering and data processing that is defined by the loaded program modules can be executed once, in the loop, or on the event. The gathered data is recorded during the execution of the module by the Logger. The collected data is transmitted to the monitoring system. There is a possibility for messages exchanging between the VM and the monitoring system. The CN Listener provides it. The monitoring system can manage the virtual machine using the Command Manager.

## DISCUSSION

The proposed method for program synthesis can be used for a number of real-life applications.

The cable TV network monitoring system is taken as a typical example of a telecommunication network. A common cable TV network contains one or more servers that serve TV receivers of the end-users. The receivers provide users a variety of services for watching TV. The task of the monitoring systems is to identify the errors that occur on the devices of the end-users, to localize them, and find their causes.

Monitoring systems are deployed on the servers, and the virtual machines are installed on TV receivers. Program modules for monitoring are synthesized on the servers and are executed by the virtual machines.

The software of the TV receivers has a multi-level structure. The software of the upper levels provides the logic of the user's services. On the low levels, the interaction with technical means is implemented. The number and composition of the intermediate levels depend on the type of receivers. The intermediate levels can include a level of the primary functions and a transport level.

The state of each user service is characterized by several parameters related to different levels of the software structure. There are links between the parameters of different levels.

The primary user services include image display services (Video), services of individual delivery of television programs and films to subscribers (VOD (Video on Demand, VOD), services for viewing private live broadcasts (PPV (Pay per View, PPV) and others. To the

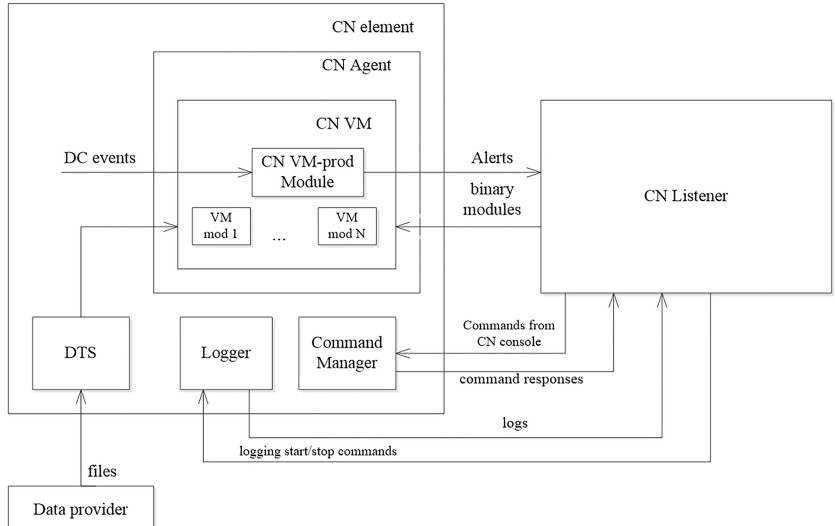

**Figure 2** Monitoring virtual machine structure.

essential services refer Electronic Program Guide service (EPG), service for writing data to disk (DVR), authorization service (Authorization), service for switching between channels (ChanelSW), etc. Transport level services allow receive/transmit data to external systems. These services receive data from the tuner (Tuning service), decode incoming data streams (Decoding service).

An error in the operation of any service results in malfunctioning of one or several end-user services.

Regular monitoring assumes control of the parameters that characterize the state of the end-user services, including Video, VOD, and PPV services. These parameters are NO_Video, NO_VOD, NO_PPV. They take the value 'true' when errors occur in the working of these services. If one of the parameters takes the value 'true', a message is formed on the TV receiver and transmitted to the monitoring system. After receiving one such message, the monitoring system synthesizes a new monitoring program for collecting data necessary to localize the error and identify its cause. The synthesized program is compiled and loaded on the receiver that has sent the message.

A model of the process of regular monitoring of a user service is presented in Fig. 3. The monitoring process may be in one of the following states: initial state S0 and final state S1, processing of a parameter S2, service is fully functional S3, service is malfunctioning S4, sending an error message about service failure S5. The values of the controlled parameters determine transitions between the states.

If an error occurs, the newly synthesized programs provide a collection of additional parameters related to services provided by the software of the lower levels. In this case, the process model has a structure that is shown in Fig. 4. The following states are defined in the model: initial state S0, final state S1, collection of an extended set of parameters S2, processing of an extended set of parameters S3, service is fully functional S4, service is

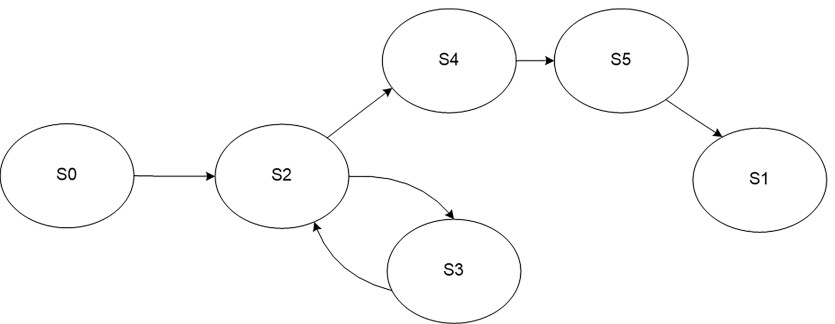

**Figure 3   Model of the regular monitoring process.**

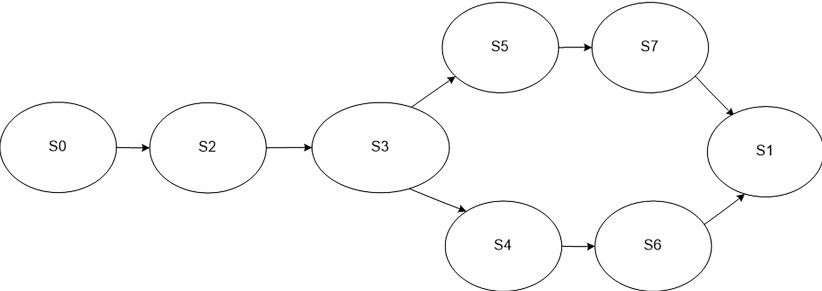

**Figure 4   Model of the monitoring process in case an error occurs in the work of a service.**

malfunctioning S5, sending an error message about the service failure S6, sending collected additional parameters S7.

Consider the situation when there is no image on the user's screen, i.e., NO_Video error has occurred. Other upper-level services, in particular, VOD and PPV, are operating normally. Thus, the initial state of the users' device is described by the following vector of the parameters {NO_Video, VOD, PPV}, the output vector is defined as {Video}. The output vector corresponds to the state of the device when there is an image on the screen. Fig. 5 shows the scheme of the synthesis that allows reaching the state {Video} from the state {NO_Video, VOD, PPV}. The synthesis is carried out in two directions - direct and reverse. Within the direct synthesis, the possibilities of transition from {NO_Video, VOD, PPV} to {Video} are considered. If such possibilities exist, then reverse synthesis allows building programs of monitoring on the base of identified possibilities.

Figure 5 shows the services of various levels and links between them. The descriptions of the software structure determine the links between the services. The solid line indicates known links that exist. The dotted lines show the links that should exist when the Video service is fully functional.

According to the Fig. 5, the result of the direct synthesis is the vector {DVR, EPG, Decoding, Authorization, ChanelSW}. The state of the Tuning service is unknown, but it is required to identify the possibility of reaching the state {Video}. Using the reverse

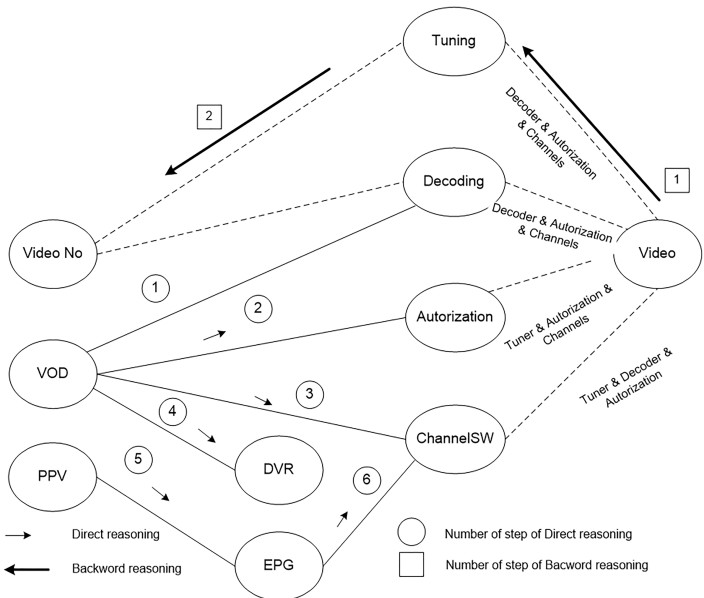

**Figure 5** The scheme of the synthesis of the monitoring programs to identify the cause of no image on users TV screens.

synthesis, the sequence {Video} -> {Tuning} -> {No_Video} is constructed. Following this sequence, it is necessary to check the tuner's function of data receiving.

A dozen parameters characterize the status of the tuner function of data received. The types of tuners determine them. The synthesized monitoring program contains commands for collecting the required set of parameters.

Below fragments of synthesized programs are presented. The following program is used for regular monitoring of the state of the Video service:

```
module main;
// event code assignment
NO_VIDEO_EVENT = < event_code> ;
start(){
// event registration
register("dc:NoVideo()",NoVideo);
}
SendAlertNoVideo(){
alertTime = _currentTime;
{
reportInt(NO_VIDEO_EVENT);
reportInt(alertTime);
return true;
}
return false;
}
```

The program below provides the collection of additional parameters that allow find the causes of no image on the users' screens:

```
module main;
start(){
// standard initialization procedures
...
//gathering required parameters
reportInt(vm_ia_firstTunerParam);
reportInt(vm_ia_secondTunerParam);
...
reportInt(vm_ia_NTunerParam);
// standard termination procedures
...
}
```

## CONCLUSION

In the article, the multilevel synthesis of programs for telecommunication networks monitoring is discussed. A new method for monitoring dynamic networks with a multilevel structure is proposed.

A new language for describing monitoring programs has been developed. The programs are translated into binary code using a compiler. The compiled programs can be executed in monitoring systems and on end-user devices. For this, a new virtual machine that is installed on the elements of the network has been designed and implemented.

The value of the proposed multilevel synthesis is that it offers the possibility to build reconfigurable monitoring systems for telecommunication networks.

The proposed method can be used for telecommunication networks monitoring in a wide range of subject domains.

Further development of the ideas of multilevel synthesis of monitoring programs assumes its application for a wide range of subject domains.

### Funding

The paper was prepared in Saint–Petersburg Electrotechnical University (LETI), and is supported by the Agreement No 075-11-2019-053 dated 20.11.2019 (Ministry of Science and Higher Education of the Russian Federation, in accordance with the Decree of the Government of the Russian Federation of April 9, 2010 No. 218), project "Creation of a domestic high-tech production of vehicle security systems based on a control mechanism and intelligent sensors, including millimeter radars in the 76-77 GHz range". The funders had no role in study design, data collection and analysis, decision to publish, or preparation of the manuscript.

## Grant Disclosures

The following grant information was disclosed by the authors:

Ministry of Science and Higher Education of the Russian Federation: 075-11-2019-053.

## Competing Interests

Andrey Kalmatskiy and Natalia Alexandrovna Zhukova were employees of Zodiac Systems, Russia & USA. Andrey Kalmatskiy is currently an employee of Google, USA.

## Author Contributions

- Man Tianxing conceived and designed the experiments, prepared figures and/or tables, authored or reviewed drafts of the paper, and approved the final draft.
- Vasiliy Yurievich Osipov, Alexander Ivanovich Vodyaho and Andrey Kalmatskiy conceived and designed the experiments, authored or reviewed drafts of the paper, and approved the final draft.
- Natalia Alexandrovna Zhukova performed the experiments, analyzed the data, prepared figures and/or tables, authored or reviewed drafts of the paper, and approved the final draft.
- Sergey Vyacheslavovich Lebedev performed the experiments, performed the computation work, prepared figures and/or tables, and approved the final draft.
- Yulia Alexandrovna Shichkina analyzed the data, performed the computation work, authored or reviewed drafts of the paper, and approved the final draft.

## Data Availability

The code is available at GitHub: https://github.com/karol11/lispy.

## Supplemental Information

Supplemental information for this article can be found online at http://dx.doi.org/10.7717/peerj-cs.288#supplemental-information.

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
