# Peer review of "Reconfigurable monitoring for telecommunication networks"

_PeerJ Computer Science, doi:10.7717/peerj-cs.288_

## Round 0.1 · original submission · Major Revisions

Please see whether you can address the serious concerns of the second reviewer: "Research does not have sufficient bases. The approach presented...is not adequately substantiated. There is no comparison with other similar methods...The article does not show the evaluation of the present approach. Were data analyzed to build the current situation? ...Compare your approach with available non/dynamic approaches - process mining, dynamic modelling and network optimization. "

·

Basic reporting

Overall language is good but with some issues though. Several examples:
Line 6: Seems -> Seem (plural form)
Line 9: of is redundant
Line: 42: medical -> medicine
The sentence in lines 57-58 is not clear and should be rephrased.
Lines 89-90 and 100: in dynamics -> dynamically

If possible, the article should be reviewed by a professional English teacher or native speaker.

Regarding other aspects, the authors provided plenty of references, background context is well described and sufficient. Generally speaking, the article is well organized and contains necessary raw data and figures. Though in the introduction authors mention sections, but it's unclear where each section starts. It'll be easier to navigate through the article if authors will add this.

Experimental design

The article meets the scope and aims of the journal. Research questions are well defined in the article introduction and then well described in the article.

Unfortunately, authors do not provide a detailed description of how to replicate their method in some end-user devices. However they provide some source code with the actual implementation of the monitoring method.

Validity of the findings

Authors provided the some examples of real-life problems that could be solved by implementing their method.

There is a clear connection between research questions, which was defined and the outcome of the work.

Reviewer 2 ·

Basic reporting

The text contains clear professional English. The list of references is sufficient. The structure of the article is appropriate. The images have a low level but sufficient informative value. The symbol table needs to be corrected.

Experimental design

The research question is meaningful but not sufficiently defined. Research does not have sufficient bases. The approach presented is exciting and original, but is not adequately substantiated. There is no comparison with other similar methods.

Validity of the findings

The article does not show the evaluation of the present approach. Were data analyzed to build the current situation? How. Compare your approach with available non/dynamic approaches - process mining, dynamic modelling and network optimization.

---

## Round 0.2 · Minor Revisions

The second reviewer comments should be addressed prior to acceptance.

·

Basic reporting

Authors addressed all my feedback

Experimental design

Authors addressed all my feedback

Validity of the findings

Authors addressed all my feedback

Additional comments

Authors addressed all my feedback

Reviewer 2 ·

Basic reporting

The presented text is not significantly different from the first version of the text. The list of references is sufficient. The structure of the article is appropriate. Images have remained poor quality, for example, figure 2 contains grey remnants of previous captions, and new captions overlap object boundaries.

Experimental design

The text still does not contain a clear justification for the choice of the method in the proposed system. The novelty of the technique and the trivial list of inappropriate ways are not sufficient conviction of the uniqueness of the approach.

Validity of the findings

The text still does not contain an analysis of the proposed approach - the main argument is the synthesis of new programs and many undefined parameters. What is the purpose and benefit of your proposal? Is it more time-efficient, has lower operating costs, set up? Does it have a lower computational complexity? How is the Entropy of the system? Have any optimizations been used? Nothing like this is discussed in the text.

Additional comments

Your approach is interesting, but you need to add a theoretical justification for its application. Convince readers about the efficiency and bonuses of your design. Perform design analyzes, compare case study results, discuss limitations and disadvantages. In this state, the text is insufficient.

---

## Round 0.3 · accepted · Accept

Please have a proofread and spell and grammar checks.

Reviewer 2 ·

Basic reporting

Thanks to the authors for editing the text. The article is now of higher quality; the text is clear and has a suitable structure; literary references are sufficient.

Experimental design

The research question and the justification for the research are already more evident.

Validity of the findings

The text now contains more information on the methods and procedures applied in this article. In the current state, the reader can already sufficiently evaluate the presented research.

Additional comments

Thanks to the authors for the revisions made and for submitting an interesting article.